biophysics/physiology/ecology

critical transitions, hysteresis, hibernation, climate, tipping points

**Author for correspondence:**
Daniel Oro
e-mail: d.oro@csic.es

# Flickering body temperature anticipates criticality in hibernation dynamics

Daniel Oro[1] and Lídia Freixas[2]

[1]Theoretical and Computational Ecology Laboratory, CEAB Center for Advanced Studies (CSIC), Acces Cala Sant Francesc 14, 17300 Blanes, Spain
[2]Granollers Natural Sciences Museum, Francesc Macià 51, 08402 Granollers, Spain

DO, 0000-0003-4782-3007

Hibernation has been selected for increasing survival in harsh climatic environments. Seasonal variability in temperature may push the body temperatures of hibernating animals across boundaries of alternative states between euthermic temperature and torpor temperature, typical of either hibernation or summer dormancy. Nowadays, wearable electronics present a promising avenue to assess the occurrence of criticality in physiological systems, such as body temperature fluctuating between attractors of activity and hibernation. For this purpose, we deployed temperature loggers on two hibernating edible dormice for an entire year and under Mediterranean climate conditions. Highly stochastic body temperatures with sudden switches over time allowed us to assess the reliability of statistical leading indicators to anticipate tipping points when approaching a critical transition. Hibernation dynamics showed flickering, a phenomenon occurring when a system rapidly moves back and forth between two alternative attractors preceding the upcoming major regime shift. Flickering of body temperature increased when the system approached bifurcations, which were also anticipated by several metric- and model-based statistical indicators. Nevertheless, some indicators did not show a pattern in their response, which suggests that their performance varies with the dynamics of the biological system studied. Gradual changes in air temperature drove transient between states of hibernation and activity, and also drove hysteresis. For hibernating animals, hysteresis may increase resilience when ending hibernation earlier than the optimal time, which may occur in regions where temperatures are sharply rising, especially during winter. Temporal changes in early indicators of critical transitions in hibernation dynamics may help to understand the effects of climate on evolutionary life histories and the plasticity of hibernating organisms to cope with shortened hibernation due to global warming.

# 1. Introduction

Evolution has selected to endorse resilience for buffering environmental impacts, protecting biological systems from failure. Complex responses to these impacts occur at all organizational levels, from cells and organs to populations and ecosystems. At the individual level, physiological responses are at the basis of resilience to enhance survival and reproduction. One particular life-history strategy that has evolved to cope with environmental stress is dormancy or torpor. Dormancy is a physiological adaptation in some plants and animals, that can remain torpid for weeks, months and even years [1,2]. In some mammals inhabiting seasonal ecosystems (such as Northern Hemisphere seasonal hibernators), dormancy during winter, also called hibernation, reduces the impacts of climatic stress. Hibernating species in those ecosystems may reduce their metabolic rate for long periods, which results in higher survival and lower fecundity than close phylogenetic species that do not hibernate [2]. The advantages of hibernating may explain the evolutionary success of ancestral mammals that survived the mass extinction at the Cretaceous–Palaeogene boundary following an unprecedented environmental perturbation [3]. A much higher than expected rate of recent extinctions in mammals has been recorded for homeothermic species, whereas hibernating species seem to cope better with environmental impacts due to anthropogenic global change [4]. Some theoretical models show that in harsh environments, hibernation may be all that allows population persistence [5].

The understanding of the physiological dynamics by which air temperature influences the seasonality of life histories is crucial to assess the resilience of hibernating species to global warming [6]. In recent times, wearable electronic loggers have allowed researchers to analyse the dynamics of physiological systems, such as body temperature fluctuating between activity and hibernation states [7,8]. A straightforward pattern emerges when examining these studies: body temperature in hibernating mammals such as squirrels, marmots, tenrecs, echidnas, dormice and hamsters, changes abruptly between these states [9–13]. However, this highly stochastic physiological system has never been explored to assess whether these sudden switches correspond to tipping points between alternative basins of attraction through critical transitions. Critical transitions are nonlinear, abrupt responses of some biological systems subjected to some type of gradual environmental stress. These transitions occur when a threshold value for resilience has been crossed due to the cumulative stress, and beyond this tipping point, there is a sudden shift of state [14,15]. Hibernation has either been studied from the perspective of resilience, which is the property that mediates transition between alternative stable states. Hibernation dynamics also appear as a promising candidate for assessing our capacity for anticipating transitions between states. The anticipation of responses to stress, especially when responses are nonlinear (e.g. critical transitions) remains a challenge for most biological systems [16,17]. Predicting the appearance of tipping points, i.e. a drastic change in the body temperature of hibernating animals following a gradual change in air temperature over the seasons, is feasible using statistical early warning signals (EWS) [18]. These EWS tools allow researchers to assess the occurrence of changes in statistical metrics just prior to critical transitions, which can be used to anticipate this nonlinear behaviour of the system.

For hibernating dynamics, critical transitions would be a particular type of transition in which a gradual change in air temperature, once past a threshold value, would trigger an overwhelming shift of body temperature between the contrasting states of hibernation and activity [15]. These contrasting states have their fluctuations since, during hibernation, all seasonal and some non-seasonal hibernators occasionally show euthermic temperatures, while during the non-hibernation season, many species show body temperature dropping below euthermic values. Despite variability in life histories evolved towards hibernation, species in seasonal environments, such as those in the Northern Hemisphere, show clear differences in average temperature between hibernation and activity periods. These differences have the potential to show tipping points and critical transitions. Other physiological systems, such as functional heterogeneity of some progenitor blood cells, show critical transitions through bifurcation thresholds [19]. Here, we deployed body temperature loggers to a small mammal, the edible dormouse (*Glis glis*), a Northern Hemisphere seasonal hibernating species, to assess the occurrence of criticality in hibernation dynamics and to deepen our understanding of the resilience of some hibernating animals to cope with environmental stress. We also assessed the performance of statistical leading indicators for anticipating a critical transition between stable states of activity in summer and hibernation in winter. Interestingly, the studied dormice are from a habitat at the edge of the distribution range, where patches of cold-temperate forests, their preferred habitat, are regressing due to rising temperatures in recent decades [20].

# 2. Methods

## 2.1. Study animals and procedures

We kept one male and one female dormouse in outdoor captivity near the area where they were born, in the southernmost zone of the Iberian population of the species (Montnegre range). Adult dormice here weigh 150–200 g. The climate in this area is Mediterranean, with average temperatures of 15.85°C (s.e. = 0.14, range: 0.47°C–34.11°C) and 800–900 mm rainfall. All individuals were 1 year old when they entered the cage. Despite being sexually mature, they did not engage in reproduction during the study. The cage was $ca$ 6 m$^3$ in size and it was set outdoors. Two nest-boxes as those used in our field study [21] were available for dormice, and food (mainly dry seeds of oak, chestnuts, hazelnuts and apple) was provided ad libitum. Dormice performed hibernation in a refuge underground specifically set for this purpose, as they mostly do in the wild. Data were collected automatically by a temperature data-logger (iButton® DS1922 L-F5, accuracy: ±0.5°C, sampling frequency: 1 h$^{-1}$). Devices were surgically implanted intraperitoneally for an almost year-long period (341 days, 8192 temperature records for each individual). We recorded air temperature at the exact same time intervals also using an iButton device.

## 2.2. Statistical analysis

To determine which parts of the time series corresponded to the activity period or the hibernation period, we ran an algorithm for calculating the iterated cumulative sums of squares (ICSS), which detects retrospective changes of variance for identifying breaking points.

We applied several metric-based and model-based approaches as leading indicators of EWS of critical transitions for changes in body temperatures during hibernation ($T_h$) and aestivation (euthermic temperature, $T_e$). We tested most indicators reviewed in [18] to assess the limitations in their application and interpretation. First, we used the 'earlywarnings' package in R for calculating metric-based indicators: BDS tests (after the initials of Brock, Dechert and Scheinkman, who originally built the test), conditional heteroskedasticity (CH), non-parametric Drift-Diffusion-Jump (DDJ) models, and generic EWS (temporal autocorrelation at lag-1, s.d. and skewness) (details on how each indicator was applied are in electronic supplementary material, appendix S1 and table S1). We performed sensitivity analyses to assess the reliability of generic EWS depending on choices for data transformation, detrending and filtering [18]. Second, we ran model-based indicators on standardized data. We began by running a potential analysis for assessing the existence of both flickering and the occurrence of two stable states in the body temperature time series. Potential analysis was performed within rolling windows of different size (ranging from 10 to half the size of the different periods considered). The plot of potential analysis shows the number of detected wells (or stable states) of the body temperature potential. We then fitted threshold AR($p$) models to identify transitions between alternative states due to flickering in the time series. By using the Kalman filter and Akaike information criterion (AIC) values, we assessed which model with different orders ($p = 1, 2, 3$) best fit the data. The models also estimated the threshold value $c$ and the variance of the process error and were as follows:

$$T(t) = \varnothing_0 + \sum_{i=1}^{p} \varnothing_i(T(t-i) - \varnothing_0) + \varepsilon(t),$$

where $T(t)$ represents the changes in body temperature over time $t$; parameters $\varnothing_i$ had two sets of values depending on $T(t-1)$ being lower or higher than the threshold value $c$; $\varepsilon(t)$ was a white noise process representing environmental variability (i.e. a Gaussian random variable with mean zero and variance $\sigma_\varepsilon^2$). We also calculated the Kendall $\tau$, which indicates the strength of the trend in the indicators for body temperatures. We also fitted time-varying AR($p$) models and compared their fit to those obtained from threshold models to confirm that the latter better described the flickering features of the body temperature time series. All AR($p$) models were fitted using the package 'setar' in R.

We also assessed the influence of air temperature on the dynamics of body temperature in our studied dormice. To simplify the analysis (i.e. avoiding including seasonality), we partitioned the time series between three periods: activity prior to hibernation, hibernation and activity after hibernation, as indicated by the breaking point analysis (electronic supplementary material, appendix S1 and table S2). The models added the air temperature covariate ($A$) into the AR($p$) models and were as follows:

$$T(t) = \beta A_i + \varnothing_0 + \sum_{i=1}^{p} \varnothing_i(T(t-i) - \varnothing_0) + \varepsilon(t).$$

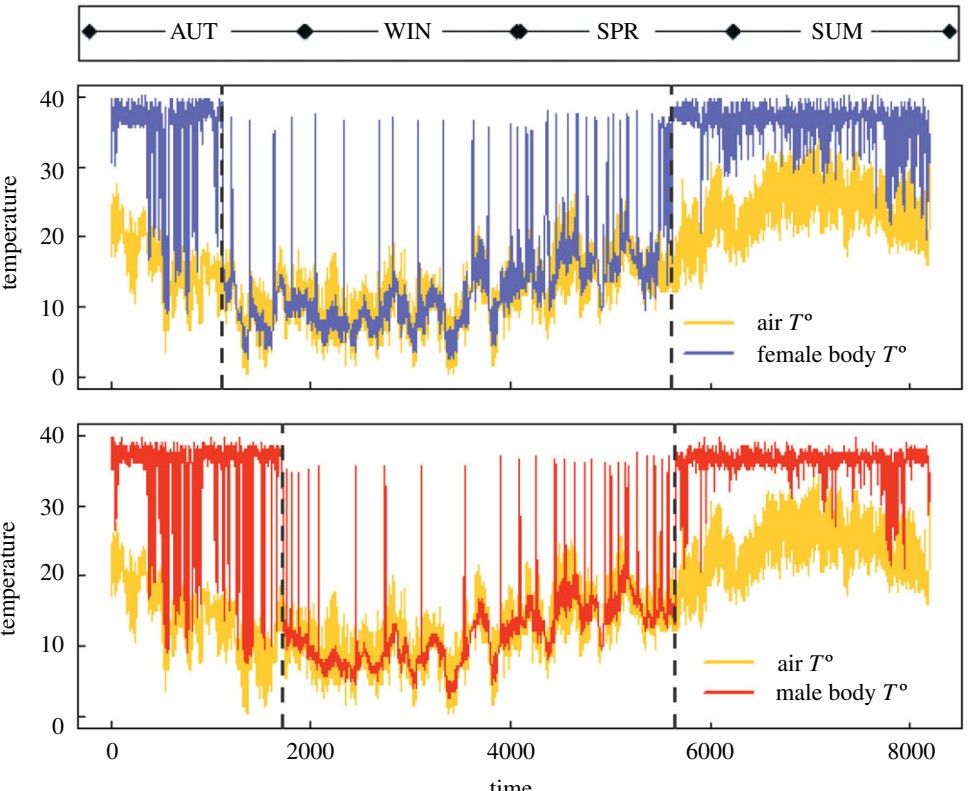

**Figure 1.** Body and air temperature recorded during the study covering an entire year (data recording began in early autumn) for the female and male dormouse (time expressed in hours). Vertical dashed lines show the two extreme breaking points (using ICSS methods) that separate the hibernation period (central part) and the activity period (external parts). Limits for each season are also shown. Flickering occurred over most of the year, especially during hibernation, and increased prior to critical transitions between states.

Here, we added the slope $\beta$ of the effect of air temperature $A_i$ on body temperature $T$. We then used AIC values of each model (with and without air temperature as explanatory variables) to select the best model. Fitting of models was carried out using the package 'TSA' in R.

# 3. Results

The time series of the dormice's body temperature suggested an increase of flickering prior to transitions, rather than a critical slowing down (figure 1). Flickering of body temperature occurred over most of the entire study period, especially during hibernation. The female entered the hibernation period 24 days earlier than the male (22 November versus 16 December, respectively), whereas the two dormice awoke from hibernation on the same day (May 29). During the activity period, $T_e$ remained rather constant (with torpor bouts starting around mid-summer), with independence of the trends in air temperature, first increasing from the termination of hibernation to mid-summer and then decreasing to the onset of the next hibernation. On the contrary, during hibernation, $T_h$ tracked air temperature very well, although several bouts to $T_e$ (as indicators of flickering) occurred with increasing frequency as the transition to activity approached (figure 1). In the following sections, we show the results of hibernation dynamics by the use of indicators developed by Dakos et al. [18].

## 3.1. Metric-based indicators

Results from BDS tests and their partial autocorrelation functions (ACF) show that we can reject the null hypothesis that the remaining time-series residuals after detrending are independent and identically distributed, which is typical of a system approaching a critical transition (electronic supplementary material, appendix S1, table S3 and figure S1). ACF also showed that during the activity period, body temperature followed a circadian cyclicity, which was not apparent during hibernation. CH was erratic and did not clearly anticipate bifurcation between states (electronic supplementary material, appendix S1 and

figure S2). DDJ metrics were noisy when plotted against time, although they were suitable indicators of resilience for flickering data: resilience decreased as conditional variance, diffusion and jump intensity increased (electronic supplementary material, appendix S1 and figure S3). Generic EWS were also noisier for the transition to hibernation, probably because the time series started just before dormice began to show an increasing frequency of torpor bouts (figure 2). In general, there was an increase in the autocorrelation at lag-1 and in variance before the two transitions, whereas skewness decreased, probably due to the increase in excursions of body temperature to $T_e$ over this period. Generic EWS showed similar performance for the time series encompassing the transitions and the whole time series (electronic supplementary material, appendix S1 and figures S4 and S5, respectively). On the contrary, air temperatures did not show abrupt changes for any of the transitions as body temperatures showed (figure 2; electronic supplementary material, appendix S1 and figure S6). The best indicators were standard deviations for the two dormice prior to enter the activity period (figure 2), but there was not a clear pattern in the behaviour of the different indicators. For instance, increase in the AR(1), which is an indicator of a critical slowing down, occurred only prior to the transition to activity for the female, but not for the transitions in the body temperature of the male nor for the transition to hibernation for the female (figure 2). Sensitivities of all generic EWS tested in our study were low, i.e. results were robust regardless of different choices on bandwidth and size of the rolling window (electronic supplementary material, appendix S1 and figure S7).

## 3.2. Model-based indicators

We fitted threshold AR($p$) models ($p$ [1,3]) and found that the best fitting model was AR(3) (figure 3). The threshold value separating the two AR(3) processes occurring at each stable state of hibernation and activity was estimated at 31°C for the two sexes. For female data, the models' fit were worse for $p = 1$ ($\Delta AIC = 1242$) and $p = 2$ ($\Delta AIC = 14$), similar to the male data ($p = 1$ ($\Delta AIC = 2630$) and $p = 2$ ($\Delta AIC = 15$)). The fit of the threshold AR(3) model was statistically better than that of a simple AR(3) model for the female ($\chi_3^2 + \chi_4^2 = 51.63$, $p < 0.001$) and the male ($\chi_4^2 + \chi_5^2 = 70.56$, $p < 0.001$). AR($p$) models also showed that air temperature explained an important part of the deviance in body temperatures, especially during hibernation, when body temperature was highly synchronous with air temperature (electronic supplementary material, appendix S1 and table S4). The fitted model showed that there were alternative states between the activity and the hibernation states, separated by unstable saddle points that corresponded to flickering temperature causing a region of bistability (figure 4). The potential landscape confirmed the occurrence of flickering and two minima (stable states) separated by a local maximum (unstable equilibrium; figure 4). The potential was lower and more narrow (lower temperature range) for $T_e$ during normal activity than for the hibernation temperature, and it showed a similar pattern for the two studied dormice. Switches between hibernation and activity occurred at different critical conditions of temperature, which was indicative of hysteresis. In late spring, dormice awoke at higher air temperatures than air temperature when they entered hibernation in late autumn (figure 5).

# 4. Discussion

We recorded data only on two individuals, but they exhibited a similar pattern of change in body temperature. Furthermore, we do not know what are the main physiological mechanisms providing resilience to the two alternative states and what causes loss of this resilience as the transition is approaching. Despite these limitations, we show that criticality occurs in the hibernation dynamics of dormice, even though air temperature changed smoothly over the year. Around fold bifurcation points, a tiny change in ambient temperature pushes body temperature across boundaries of alternative states (from a euthermic temperature to a hibernation temperature) through large transitions. These critical transitions are characterized by flickering and punctuated changes in body temperature. Flickering is a type of complex dynamics that occur when a system rapidly moves back and forth to the vicinity of an alternative state preceding the upcoming critical transition [16,22]. In our study, flickering of body temperature started far from bifurcation points and it increased when the system approached these points. Whether these dynamics occur beyond seasonal physiological hibernators, e.g. in species showing much shorter shifts between euthermic and torpid states during all seasons, remains to be explored. We show that flickering anticipated critical transitions in hibernating dormice, as it probably does for dormice from the core of the range distribution and for other seasonal hibernating mammals in the Northern Hemisphere, when looking at the dynamics of

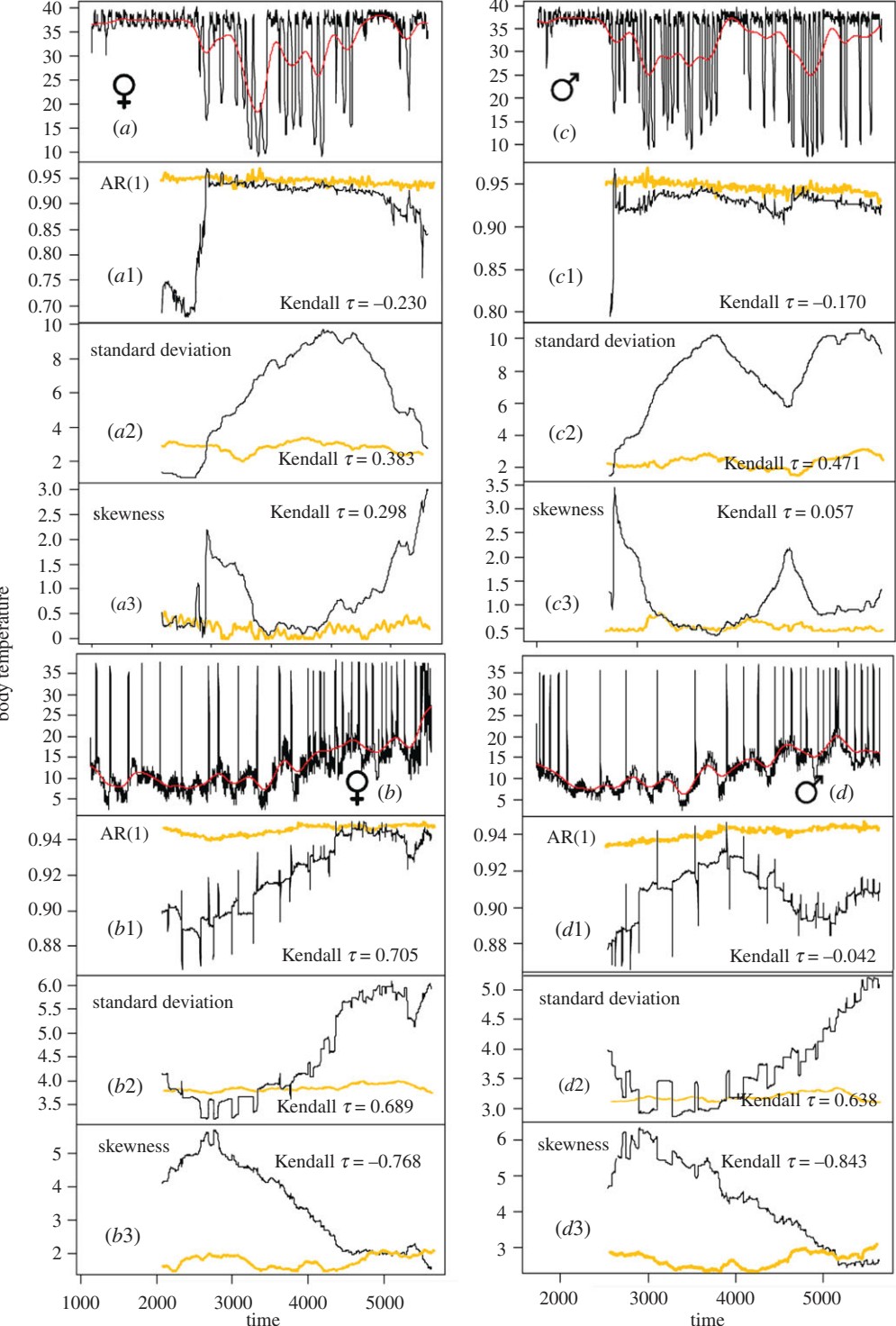

**Figure 2.** Metric-based rolling window indicators estimated for dormice body temperature (black lines) and air temperature (yellow lines) separated for each alternative state (activity and hibernation). (a,b) Activity state and hibernation state for the female; (c,d) Activity state and hibernation state for the male. Red lines show the Gaussian filtering of the time series, which dampens the main noise of the time series. Panels 1, 2 and 3 show autocorrelation at lag-1 (AR(1)), standard deviation and skewness respectively, estimated within sliding windows of 20% the size of the time series. Yellow lines show the indicators for air temperature. The Kendall $\tau$ indicates the strength of the trend in the indicators for body temperatures.

body temperatures [9,10,12]. In humans, flickering may also anticipate physiological responses, such as epileptic seizures and narcolepsy [15,23]. Together with flickering, other indicators of EWS anticipated the approaching of critical transitions in hibernation dynamics. Previous studies suggest that, contrary

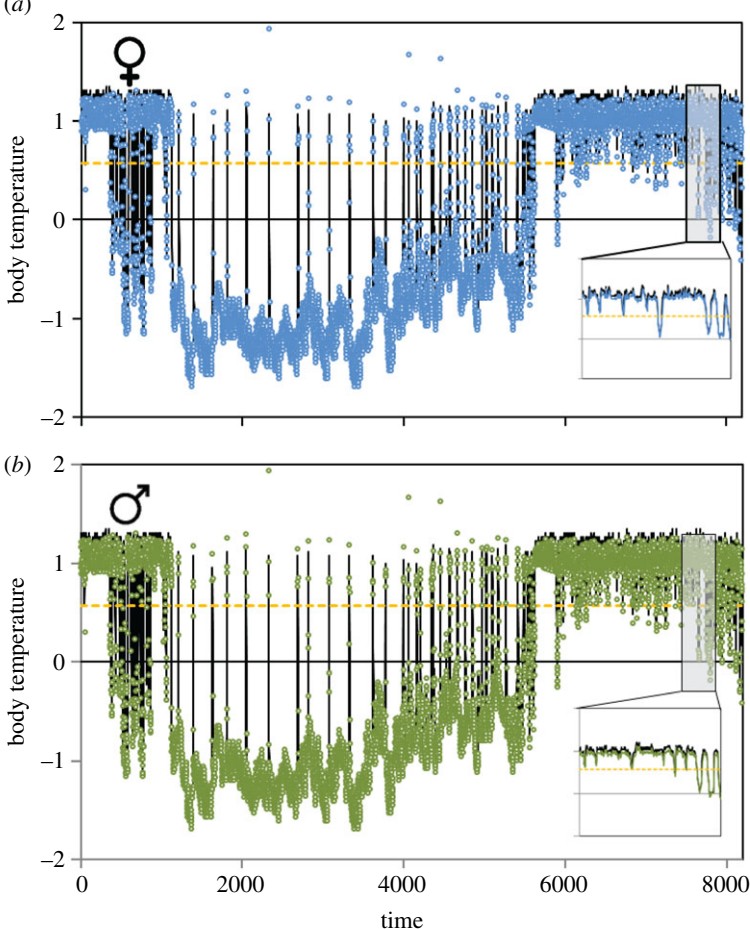

**Figure 3.** Fit of a threshold AR(3) model to body temperature (standardized data) for the female and the male dormice (a and b, respectively). Blue and green points result from the fitted AR(3) model (for female and male, respectively) and black lines show the real data. The yellow lines show the threshold value that separates the two AR(3) processes (which equal 31℃ for the two sexes). The inner panels show a zoom of an arbitrary chosen subset of the original dataset shown by the shaded area.

to critical slowing down, flickering is associated with an increase in both variance and lag-1 autocorrelation, as we found in our study [22,24]. Nevertheless, the rest of the indicators were less informative and even erratic: for instance, CH did not anticipate bifurcation, while non-parametric DDJ metrics were very noisy. As Dakos *et al.* [18] pointed out when developing the indicators, the performance of any indicator, as well as the interpretations based on them, is likely depending on the features and dynamics of the biological system studied.

Hibernation dynamics of dormice also have hysteresis, which shows the tendency of body temperature to remain on the same attractor until attaining a critical threshold value of air temperature, with the particularity that bifurcations occurred at different air temperatures for each of the two cyclic transitions. These cyclic transitions resemble the dynamics of wake–sleep and micro-sleeps occurring for circadian cycles in mammals [23]. Since hibernation is driven by seasonal climate, its dynamics follow cycles that are locked into phase [15]. This type of complex cyclic dynamics occur between coupled oscillators and commonly occurs in nature (e.g. heart beating, reproductive events and predator–prey fluctuations). Epileptic seizures mentioned above occur by the phase locking of firing in neural cells [16]. In our study, locking of forced hibernation by forcing winter occurs with 1 : 1 rhythm, which means that little climate forcing is enough for locking [15]. Phase locking occurs for global climatic indexes such as NAO and ENSO, which are known to be coupled with several local seasonal ecological processes.

Statistical indicators of EWS quantify critical transitions and resilience for very different ecological systems [25], and we show that these generic indicators can be also applied to other dynamical systems sharing their fundamental properties, such as physiological hibernation. Other physiologically critical transitions during hibernation dynamics may occur and may be anticipated, such as lipid structure and enzyme function of mitochondrial membranes from the liver, kidney, brown fat and

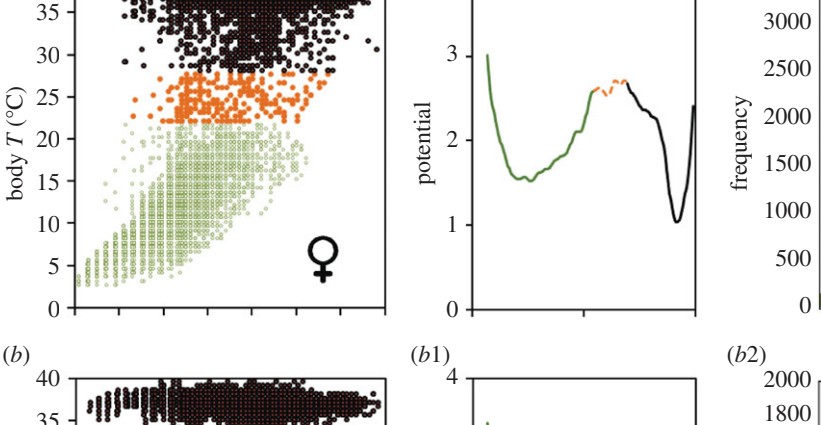

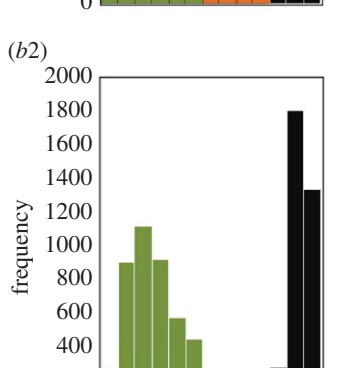

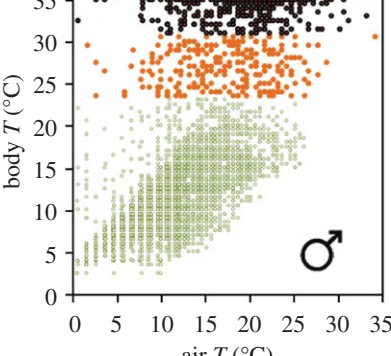

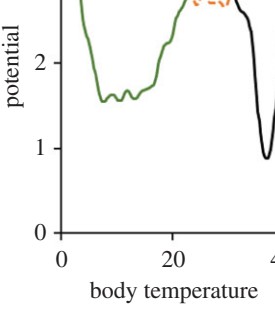

**Figure 4.** Left panels: dormice's body temperature as a function of air temperature variability (*a* and *b* for female and male, respectively). Central panels (*a*1 and *b*1) show the potential for body temperature for female and male dormouse, respectively, whereas right panels (*a*2 and *b*2) show the bimodal distribution of body temperatures around the ones characterizing each of the two alternate states: the hibernation state (green colours) and the activity state (black colours), separated by an unstable state (orange colours). The entire time series was used here.

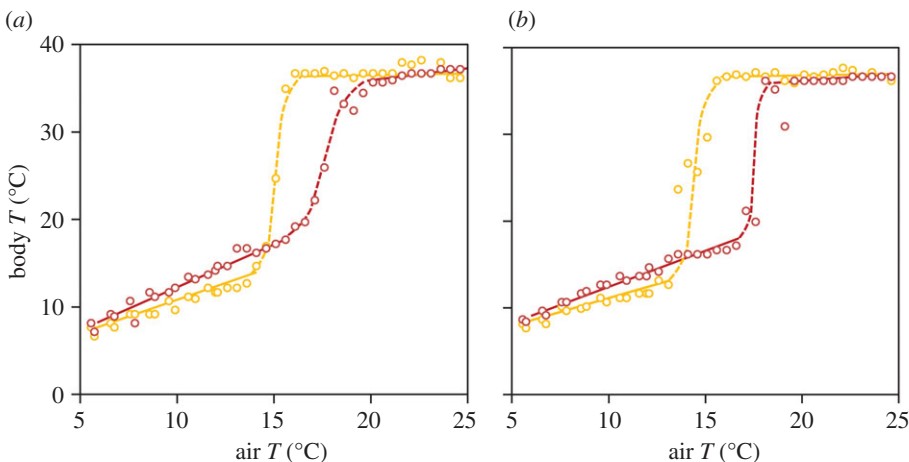

**Figure 5.** Hysteresis occurring between hibernation and activity states for the studied dormice (*a* and *b* correspond to female and male, respectively). Orange lines and points show the transition to hibernation (body temperature drop when air temperature decreased below a critical point) and red lines and points show the transition to an euthermic, activity state (body temperature increase when air temperature crossed a critical point). Each point represents the median value of body temperatures for each air temperature.

heart tissues. Most physiological systems are high-dimensional, meaning that they are not instantaneous, they show temporal delays and these delays may drive transient dynamics [26]. Examples of high-dimensional systems with high degrees of freedom and non-gradual responses are blood cells with

functional heterogeneity or neurons involved in sleep–wake cycles, which also show critical transitions that can be predicted before bifurcation [19]. The same dynamics occur for homeostatic changes in hormone regulation, immune responses, gene expression and asthma incidence [27,28]. Our capacity to anticipate pathological changes and loss of resilience, e.g. due to opposite physiological commitment to that intended in normal conditions, is crucial for human health [7]. Interestingly, hibernation dynamics and human health converge due to the increased scientific interest in the benefits of dormancy for humans in coping with different stresses, e.g. the potential of a hibernating state in astronauts for deep space travel [3]. Similarly, EWS can be used to detect changes in the nonlinear dynamics of body temperature and the resilience of hibernating mammals due to increasing climatic stress. Physiological variables can perform better to anticipate nonlinear dynamics than noisier ecological variables [29,30], and the former may be a good bio-indicator of ecological changes and temporal variability in resilience. Given the potential consequences that hibernation dynamics may have on population fluctuations and extinction, there is a growing concern about the impacts of climate warming on hibernating species [31]. This is especially true for extreme climatic events and the capacity of organisms for climate resilience in evolutionary life-history strategies [32,33].

The influence of climate may be greater for species such as edible dormice in our Mediterranean study area. Here, heat and drought waves are associated and their frequency is increasing [34,35], which is affecting cold-temperate forests, the preferred habitat of dormice [20]. The potential impact on hibernating mammals in many regions is likely to be related to a loss of suitable habitat, mediated by climate variability and not to a direct impact on hibernation dynamics. This is because hibernation is a very effective resilient physiological mechanism to cope with climatic stress [36–38]. Animals can adjust hibernation to maximize fitness because it influences life-history traits such as recruitment by age and the onset of reproduction [39]. The evolution of hibernation has selected for a very plastic trait. The duration of hibernation may change with seasonal climate variability and with the availability of resources. For instance, rising temperatures cause earlier emergence from hibernation in the yellow-bellied marmot (*Marmota flaviventris*), which has led to a longer growing season and larger body masses before entering hibernation [40]. The demographic consequences included higher adult survival and a sharp increase in population growth rate. Hibernation in dormice shows large differences between populations depending on local climate, and harsher and longer winter means longer hibernation times, e.g. up to 11 months [12]. Interestingly, life-history strategies of dormice are very plastic depending on those local climatic conditions, since as long as the duration of hibernation increases, adult survival is higher and fertility is lower [36,41,42]. Hysteresis, such as that shown by the studied Mediterranean edible dormice, may also increase resilience by avoiding the termination of hibernation earlier than the optimal time, which may occur in regions where temperatures are sharply rising, especially during winter [38]. Experimental studies looking at how environmental stress affects hibernation dynamics (e.g. [43]) are promising to assess the changes in leading indicators of critical transitions. Exploring how criticality and tipping points appear in different hibernating animals with different evolutionary life histories and with varying ecological features may also shed light on how the resilience of physiological systems cope with environmental stress.

# 5. Conclusion

Nonlinear dynamics, such as critical transitions, commonly occur in nature. However, anticipating such transitions remains a challenge and it hinders our understanding of dynamical systems, such as physiological responses to environmental changes. Hibernation evolved as a physiological strategy to cope with air temperature stress. Here, we show that temporal variability in the body temperature of a hibernating mammal is at criticality. Transitions to both hibernation and activity were anticipated by flickering, a phenomenon occurring when a system rapidly shifts between alternative basins of attraction close to bifurcations. Hibernation dynamics showed hysteresis and critical transitions were driven by gradual changes in air temperature. The understanding of hibernation dynamics is important to assess the role of physiological resilience to climatic stress, population dynamics and population extinctions.

Ethics. All research was permitted by the Government of Catalonia, License 014/2009. This research was reviewed and approved by DARP (Generalitat of Catalonia), following Decree 214/1997, of 30 July, which regulates the use of animals for experimentation and other scientific purposes.

Data accessibility. All data used in the analyses are available from CSIC Repository: http://hdl.handle.net/10261/194623.

Authors' contributions. D.O. and L.F. conceived the study; L.F. performed the experimental work; D.O. carried out the statistical analysis and the writing.

Competing interests. We declare we have no competing interests.

Funding. This work was carried out without any specific funding.

Acknowledgements. We are grateful to Claudia Bieber and Gabrielle Stalder for technical advice, to Andrea Chirifre, Javier Millan and Jordi Grífols for conducting veterinarian work, to Antoni Arrizabalaga, Pilar Saborit and Silvia Miguez for logistic support and to Matthieu Stigler and Timothy Cline for helping with R packages. Two anonymous reviewers provided very helpful comments for improving the MS.

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
