## [Reviewer comments · Royal Society Open Science]

Review History

RSOS-201571.R0 (Original submission)

Review form: Reviewer 1

Is the manuscript scientifically sound in its present form?

Yes

Are the interpretations and conclusions justified by the results?

Yes

Is the language acceptable?

Yes

Do you have any ethical concerns with this paper?

No

Have you any concerns about statistical analyses in this paper?

No

Recommendation?

Accept with minor revision (please list in comments)

Comments to the Author(s)

This work analyzes the annual cycle of body temperature of the edible dormouse. The study uses sound statistical analysis to show that the transitions to the hibernation period and the converse transition to active live in Spring can be regarded as critical transitions between two alternate resilient (stable) states. In particular, this work shows that such observed transitions have common characteristics (in statistical sense, such as flickering) with the critical transitions close to tipping points between alternate states that have been previously described with the use of mathematical models in other natural systems.

I believe the ms provides a nice example of flickering preceding both forward and backward abrupt transitions between two alternate states. Empirical data also show hysteresis.

The paper uses well-developed statistical tests applied to time series data to detect early warning signals of abrupt transitions. The authors use both metric-based and model-based approaches to detect early warning signals. Model-based methods are not based on a dynamical system where an attempt to include some physiological mechanisms responsible for this alternate behavior could have been explored, but only a statistical time series model.

The work considers outdoor temperature as the control variable, which smoothly changes along the year, driving an abrupt change in the response variable, the body temperature. The main drawback of this work is that there is little discussion about the main physiological mechanisms providing resilience to both alternate states and what causes the erosion of this resilience as the transition is close. This could have been explore with a dynamical systems of some sort. The paper useses concepts from dynamical systems, but then only uses a statistical time series model. I believe this exploration was not the goal of the work presented here by these authors. However, some speculative comments in the discussion could have been added.

One question to further explore is whether or not there are signs of flickering in air temperature as well. One can imagine that spring and fall are transition seasons (with more temperature fluctuations close to the end of the season) between two alternate states, Summer and Winter.

Finally, I find Figure 4 central to the paper. I would have added the histogram (as a third columns on the left) showing a bimodal distribution of body temperatures around the ones characterizing each of the alternate states.

Review form: Reviewer 2

Is the manuscript scientifically sound in its present form?

No

Are the interpretations and conclusions justified by the results?

No

Is the language acceptable?

Yes

Do you have any ethical concerns with this paper?

No

Have you any concerns about statistical analyses in this paper?

No

Recommendation?

Major revision is needed (please make suggestions in comments)

Comments to the Author(s)

The authors are providing analyses on the time dynamics of body temperature (T°) in two individuals of the species *Glis glis* in the Mediterranean climate. They use one annual time series of this measure and study critical transitions related to the entrance and exit from hibernation. The authors use several metric-based techniques to identify critical transitions from time series analyses, based on previous literature of Early Warning Signals (EWS, leading indicators). They also use model-based indicators, using e.g., threshold AR(p) models.

In my opinion the authors follow an interesting approach to provide EWS by monitoring an in vivo system. This kind of research is not easy and they did an effort in doing so. They have based on dynamical behaviour with flickering body T° . They have only used two individuals to carry out the analyses, but they show a common pattern with some differences into the entry of the hibernation period. The study would have benefit from using more individuals, not to perform all of the analysis to each of them, but to see whether the signals they find for the body T° remain similar between individuals (or even between ages). The use of more individuals could have been used to extract a common pattern of changes between hibernation and activity, and see the intrinsic variability associated to it under similar patterns of air T° . I pretty much enjoyed the Discussion Section, which is very well written and characterises quite well the topic and work done by the authors. However, a major problem is that the Results Section is very short and poor. The article is well written and novel. However, I have some concerns that should be addressed in a Major revision. Some of these concerns are about the solid evidences they are providing to show that these transitions may occur, especially in the transition towards hibernation.

Major concerns:

The detection of early warning signals (EWS) is easier when the control parameter varies smoothly and the system undergoes an abrupt transition, but the time series are very noisy (air T° is extremely noisy). I have also some questions regarding the experiments and the presentation of some of the results. My main concern is that from the analyses performed (mainly the metric-based measures) I do not see a clear matching between some of the changes in these indicators and the transitions identified in the data (marked with vertical dashed lines in Fig. 1). This is specially important for the entry to hibernation, where the time series starting at time 0 up to the first vertical line is very short. I understand that it might have took longer time to do the experiments, but it would have been better to have a two-year time series, as a way to have "temporal" replicates of the transitions for the same individuals. Regarding this concern, for example, the female shows a clear flickering pattern approximately in the middle of the first period (beginning Autumn) towards the first transition, but close to the transition the flickering seems to disappear. For the male this phenomenon is not so evident. In this sense, I am wondering if the title of this article is: "Flickering body temperature anticipates criticality in hibernation dynamics" may be changed. From the words in the title I understand that the main result of the paper is an evident warning signal prior to the entry into hibernation. However, as I previously stated, from the data of the time series shown in Fig. 1, it is not clear to me that the entry into hibernation for the female records a special, and high-frequency, flickering pattern.

The quality of the figures is not very good, and, for example, it is difficult to see the data of the air T° in Fig. 1. I would suggest the authors to use high quality figures along the article. I suggest to

use another colour (perhaps light grey with some transparency) for the air T° and overlap it to the body T° , and also show the air T° separately. That is, the authors should plot the air T° of the two panels in Fig. 1 alone and separately, perhaps as a Supplementary figure.

Despite a visual inspection of the body temperature shows the increase in flickering frequency close to the vertical dashed lines of Fig. 1, there exists flickering during all the hibernation period. This means, I guess, that mice have some activity during the hibernation period that becomes more frequent as spring approaches. If this is the case, it should be better explained in the article and in the caption of Fig. 1.

I understand that the body T° was monitored every hour, thus they are not continuous data points. How could have this affected in detecting activity periods shorter than 1 hour? If the time axes in the plots are hours, just write hours.

The results Section is extremely short. This section should be worked more (see below).

Lines 176-177: the ACF shown in Fig. S1 is large only at lag-1. Makes it sense close to a critical transition? I would expect correlations at several lags.

Figure 2. Panels (a) and (b) are body temperature? Before hibernation, right? Please put Temperature in the y-axis. Why did you applied a Gaussian filtering to the time series? Concerning the period from time = 0 to time = 1800 (approx) - entry into hibernation: The standard deviation (SD) for the female shows a peak at about time = 8000, while the change to the other "hibernation attractor" takes place at about time = 1800. This difference makes me to doubt that this is indicating the presence of a critical transition for this period. Also, the SD for the male shows two peaks, thus one could think there are two critical points here (which is not the case, right?). The skewness for the female seems OK, this is increasing. However, the skewness for the mail does not show a clear pattern. You should discuss more this result, and in general, you should discuss more deeply Fig. 2 to convince the reader about your conclusions. Concerning the period from time = 2000 to time = 5800 (approx) - approach to activity and exit from hibernation: The results for the SD are here clearer. However, why is the skewness decreasing? Beyond a more extensive discussion and explanations of the results (including the Suppl. Mat.) I may suggest to include a paragraph at the end of the results Section summarising the EWS indicators confirming the presence of this critical transitions the authors claim to exist. That is, to name the measures allowing to obtain a clear result about the approach to a critical transition.

Figure 3. What does this 31°C threshold for the AR(3) model mean?

Figure 4. Panels (a) and (b) look nice. Here, the two stable states seem more or less clear from the data. I have some questions: have you computed this type of bifurcation diagram with the fitted AR(3) model? If so, you should put this info in the caption of fig. 4. Have you tried to build this diagram using the raw T° data? I would like to see it, perhaps it looks to noisy, but if you plot a hot map with the density of points within given regions you could also observe this "fold"-like behaviour.

How have the solid lines been placed within this cloud of points? Just visually? Or you did some analyses to make them pass through the more dense cloud of points? I am mentioning it because the lower branches seems ok to me, but the upper ones are missing lots of points at the left part of the plots. The same for the bifurcation points, have been computed or they have drawn visually? How have the potentials of panels (a1) and (b1) been computed?

Figure 5: Why do you talk about hysteresis here? I do not see how you relate hysteresis with the shape of these curves. This plot is very difficult to understand. You are showing two curves (and points), one for hibernation and the other for the euthermic state. But both have the same shape. These results are very difficult to interpret.

Table S1. Generic EWS: The authors used an overlapping moving window, but, could this affect the autocorrelation measures? They need to justify why not if this is not the case.

Table S4. The Standard errors are shown by means of the standard deviation, de typical deviation, or the standard error of the mean? These error for parameter ϕ_0 are too large to take conclusions. Where is (A) in the Table?

Figure S6. The sensitivities of the generic EWS seem quite heterogeneous and changing to the rolling window metrics. With such a variability I wonder if it is possible to get conclusions from the rolling windows used in the analyses. Am I missing anything?

Minor concerns:

The concept of Early Warning Signals (EWS) should be explained in the Introduction.

Line 77: critical transitions can also be continuous.

Line 130: Indicate what does BDS means

Figure 2. The measures performed in each panel should go in the y-axes. Could you add the symbols of male and female to the panels to distinguish them better? You can do it throughout the paper to ease the interpretation of results for each sex.

Line 228: Please, write Early Warning Signals (EWS) the first time they appear in the Discussion (or, generically, in a given section).

Lines 232-233: Please, indicate what CH and DDJ mean.

Line 255: What does high-dimensional physiological systems mean?

Lines 269-271: I think it would be interesting to extend the comments on the consequences of hibernation on population fluctuations and especially, on species extinctions.

Line 306-307: The authors say that flickering occurs when a system rapidly shifts between alternative basins of attraction far from bifurcations. Far or close to bifurcations?

Decision letter (RSOS-201571.R0)

Dear Professor Oro

The Editors assigned to your paper RSOS-201571 "Flickering body temperature anticipates criticality in hibernation dynamics" have now received comments from reviewers and would like

you to revise the paper in accordance with the reviewer comments and any comments from the Editors. Please note this decision does not guarantee eventual acceptance.

Please submit your revised manuscript and required files (see below) no later than 21 days from today's (ie 20-Nov-2020) date. Note: the ScholarOne system will 'lock' if submission of the revision is attempted 21 or more days after the deadline. If you do not think you will be able to meet this deadline please contact the editorial office immediately.

on behalf of Dr Cynthia Downs (Associate Editor) and Pete Smith (Subject Editor)
openscience@royalsociety.org

Associate Editor Comments to Author (Dr Cynthia Downs):

Associate Editor: 1

Comments to the Author:

Thank you for submitting to Open Science. Two reviewers and I reviewed this manuscript. This manuscript presents an analysis of the time dynamics of body temperature in two individuals (one male and one female) of Edible dormouse (*Glis glis*). The research shows that the two individuals exhibit a similar pattern of flickering body temperature at the transition between two stable temperature states (hibernation and active). One reviewer provided a favorable review with minimal comments, and one provided more substantial feedback. Reviewer 2 expressed concern that the study used only two dormice and a single year of data. This reviewer suggested that the work could have benefited from a second annual year to better support body temperature patterns entering fall. Although I agree that adding additional individuals would broaden the results' scope, I do not think that this is necessary for publication. Instead, I recommend adding a discussion of the study's limitations that arise from the study design (e.g., sample size and rate of temperature recordings). Additionally, I encourage you to address the

comments about the figures and table to help clarify the results and conclusions, particularly figure 4, which is critical to the paper. Specifically, for Fig 4, please add tick marks at the 5's on the y-axis to make it easier to compare body and air temperature.

Reviewer comments to Author:

Reviewer: 1

Comments to the Author(s)

This work analyzes the annual cycle of body temperature of the edible dormouse. The study uses sound statistical analysis to show that the transitions to the hibernation period and the converse transition to active live in Spring can be regarded as critical transitions between two alternate resilient (stable) states. In particular, this work shows that such observed transitions have common characteristics (in statistical sense, such as flickering) with the critical transitions close to tipping points between alternate states that have been previously described with the use of mathematical models in other natural systems.

I believe the ms provides a nice example of flickering preceding both forward and backward abrupt transitions between two alternate states. Empirical data also show hysteresis.

The paper uses well-developed statistical tests applied to time series data to detect early warning signals of abrupt transitions. The authors use both metric-based and model-based approaches to detect early warning signals. Model-based methods are not based on a dynamical system where an attempt to include some physiological mechanisms responsible for this alternate behavior could have been explored, but only a statistical time series model.

The work considers outdoor temperature as the control variable, which smoothly changes along the year, driving an abrupt change in the response variable, the body temperature. The main drawback of this work is that there is little discussion about the main physiological mechanisms providing resilience to both alternate states and what causes the erosion of this resilience as the transition is close. This could have been explore with a dynamical systems of some sort. The paper uses concepts from dynamical systems, but then only uses a statistical time series model. I believe this exploration was not the goal of the work presented here by these authors. However, some speculative comments in the discussion could have been added.

One question to further explore is whether or not there are signs of flickering in air temperature as well. One can imagine that spring and fall are transition seasons (with more temperature fluctuations close to the end of the season) between two alternate states, Summer and Winter.

Finally, I find Figure 4 central to the paper. I would have added the histogram (as a third columns on the left) showing a bimodal distribution of body temperatures around the ones characterizing each of the alternate states.

Reviewer: 2

Comments to the Author(s)

The authors are providing analyses on the time dynamics of body temperature (T^b) in two individuals of the species *Glis glis* in the Mediterranean climate. They use one annual time series of this measure and study critical transitions related to the entrance and exit from hibernation. The authors use several metric-based techniques to identify critical transitions from time series analyses, based on previous literature of Early Warning Signals (EWS, leading indicators). They also use model-based indicators, using e.g., threshold AR(p) models.

In my opinion the authors follow an interesting approach to provide EWS by monitoring an in vivo system. This kind of research is not easy and they did an effort in doing so. They have based on dynamical behaviour with flickering body T° . They have only used two individuals to carry out the analyses, but they show a common pattern with some differences into the entry of the hibernation period. The study would have benefit from using more individuals, not to perform all of the analysis to each of them, but to see whether the signals they find for the body T° remain similar between individuals (or even between ages). The use of more individuals could have been used to extract a common pattern of changes between hibernation and activity, and see the intrinsic variability associated to it under similar patterns of air T° . I pretty much enjoyed the Discussion Section, which is very well written and characterises quite well the topic and work done by the authors. However, a major problem is that the Results Section is very short and poor. The article is well written and novel. However, I have some concerns that should be addressed in a Major revision. Some of these concerns are about the solid evidences they are providing to show that these transitions may occur, especially in the transition towards hibernation.

Major concerns:

The detection of early warning signals (EWS) is easier when the control parameter varies smoothly and the system undergoes an abrupt transition, but the time series are very noisy (air T° is extremely noisy). I have also some questions regarding the experiments and the presentation of some of the results. My main concern is that from the analyses performed (mainly the metric-based measures) I do not see a clear matching between some of the changes in these indicators and the transitions identified in the data (marked with vertical dashed lines in Fig. 1). This is specially important for the entry to hibernation, where the time series starting at time 0 up to the first vertical line is very short. I understand that it might have took longer time to do the experiments, but it would have been better to have a two-year time series, as a way to have "temporal" replicates of the transitions for the same individuals. Regarding this concern, for example, the female shows a clear flickering pattern approximately in the middle of the first period (beginning Autumn) towards the first transition, but close to the transition the flickering seems to disappear. For the male this phenomenon is not so evident. In this sense, I am wondering if the title of this article is: "Flickering body temperature anticipates criticality in hibernation dynamics" may be changed. From the words in the title I understand that the main result of the paper is an evident warning signal prior to the entry into hibernation. However, as I previously stated, from the data of the time series shown in Fig. 1, it is not clear to me that the entry into hibernation for the female records a special, and high-frequency, flickering pattern.

The quality of the figures is not very good, and, for example, it is difficult to see the data of the air T° in Fig. 1. I would suggest the authors to use high quality figures along the article. I suggest to use another colour (perhaps light grey with some transparency) for the air T° and overlap it to the body T° , and also show the air T° separately. That is, the authors should plot the air T° of the two panels in Fig. 1 alone and separately, perhaps as a Supplementary figure.

Despite a visual inspection of the body temperature shows the increase in flickering frequency close to the vertical dashed lines of Fig. 1, there exists flickering during all the hibernation period. This means, I guess, that mice have some activity during the hibernation period that becomes more frequent as spring approaches. If this is the case, it should be better explained in the article and in the caption of Fig. 1.

I understand that the body T° was monitored every hour, thus they are not continuous data points. How could have this affected in detecting activity periods shorter than 1 hour? If the time axes in the plots are hours, just write hours.

The results Section is extremely short. This section should be worked more (see below).

Lines 176-177: the ACF shown in Fig. S1 is large only at lag-1. Makes it sense close to a critical transition? I would expect correlations at several lags.

Figure 2. Panels (a) and (b) are body temperature? Before hibernation, right? Please put Temperature in the y-axis. Why did you applied a Gaussian filtering to the time series? Concerning the period from time = 0 to time = 1800 (approx) - entry into hibernation: The standard deviation (SD) for the female shows a peak at about time = 8000, while the change to the other "hibernation attractor" takes place at about time = 1800. This difference makes me to doubt that this is indicating the presence of a critical transition for this period. Also, the SD for the male shows two peaks, thus one could think there are two critical points here (which is not the case, right?). The skewness for the female seems OK, this is increasing. However, the skewness for the mail does not show a clear pattern. You should discuss more this result, and in general, you should discuss more deeply Fig. 2 to convince the reader about your conclusions. Concerning the period from time = 2000 to time = 5800 (approx) - approach to activity and exit from hibernation: The results for the SD are here clearer. However, why is the skewness decreasing? Beyond a more extensive discussion and explanations of the results (including the Suppl. Mat.) I may suggest to include a paragraph at the end of the results Section summarising the EWS indicators confirming the presence of this critical transitions the authors claim to exist. That is, to name the measures allowing to obtain a clear result about the approach to a critical transition.

Figure 3. What does this 31°C threshold for the AR(3) model mean?

Figure 4. Panels (a) and (b) look nice. Here, the two stable states seem more or less clear from the data. I have some questions: have you computed this type of bifurcation diagram with the fitted AR(3) model? If so, you should put this info in the caption of fig. 4. Have you tried to build this diagram using the raw T^o data? I would like to see it, perhaps it looks to noisy, but if you plot a hot map with the density of points within given regions you could also observe this "fold"-like behaviour.

How have the solid lines been placed within this cloud of points? Just visually? Or you did some analyses to make them pass through the more dense cloud of points? I am mentioning it because the lower branches seems ok to me, but the upper ones are missing lots of points at the left part of the plots. The same for the bifurcation points, have been computed or they have drawn visually? How have the potentials of panels (a1) and (b1) been computed?

Figure 5: Why do you talk about hysteresis here? I do not see how you relate hysteresis with the shape of these curves. This plot is very difficult to understand. You are showing two curves (and points), one for hibernation and the other for the euthermic state. But both have the same shape. These results are very difficult to interpret.

Table S1. Generic EWS: The authors used an overlapping moving window, but, could this affect the autocorrelation measures? They need to justify why not if this is not the case.

Table S4. The Standard errors are shown by means of the standard deviation, de typical deviation, or the standard error of the mean? These error for parameter ϕ_0 are too large to take conclusions. Where is (A) in the Table?

Figure S6. The sensitivities of the generic EWS seem quite heterogeneous and changing to the rolling window metrics. With such a variability I wonder if it is possible to get conclusions from the rolling windows used in the analyses. Am I missing anything?

Minor concerns:

The concept of Early Warning Signals (EWS) should be explained in the Introduction.

Line 77: critical transitions can also be continuous.

Line 130: Indicate what does BDS means

Figure 2. The measures performed in each panel should go in the y-axes. Could you add the symbols of male and female to the panels to distinguish them better? You can do it throughout the paper to ease the interpretation of results for each sex.

Line 228: Please, write Early Warning Signals (EWS) the first time they appear in the Discussion (or, generically, in a given section).

Lines 232-233: Please, indicate what CH and DDJ mean.

Line 255: What does high-dimensional physiological systems mean?

Lines 269-271: I think it would be interesting to extend the comments on the consequences of hibernation on population fluctuations and especially, on species extinctions.

Line 306-307: The authors say that flickering occurs when a system rapidly shifts between alternative basins of attraction far from bifurcations. Far or close to bifurcations?

===PREPARING YOUR MANUSCRIPT===

Your revised paper should include the changes requested by the referees and Editors of your manuscript. You should provide two versions of this manuscript and both versions must be provided in an editable format:
 one version identifying all the changes that have been made (for instance, in coloured highlight, in bold text, or tracked changes);
 a 'clean' version of the new manuscript that incorporates the changes made, but does not highlight them. This version will be used for typesetting if your manuscript is accepted.
 Please ensure that any equations included in the paper are editable text and not embedded images.

If you have been asked to revise the written English in your submission as a condition of publication, you must do so, and you are expected to provide evidence that you have received language editing support. The journal would prefer that you use a professional language editing service and provide a certificate of editing, but a signed letter from a colleague who is a native

speaker of English is acceptable. Note the journal has arranged a number of discounts for authors using professional language editing services (<https://royalsociety.org/journals/authors/benefits/language-editing/>).

===PREPARING YOUR REVISION IN SCHOLARONE===

<https://royalsociety.org/journals/authors/author-guidelines/#supplementary-material> to include a suitable title and informative caption. An example of appropriate titling and captioning may be found at https://figshare.com/articles/Table_S2_from_ls_there_a_trade-

off_between_peak_performance_and_performance_breadth_across_temperatures_for_aerobic_sc
ope_in_teleost_fishes_/3843624.

Author's Response to Decision Letter for (RSOS-201571.R0)

See Appendix A.

Decision letter (RSOS-201571.R1)

Dear Professor Oro,

It is a pleasure to accept your manuscript entitled "Flickering body temperature anticipates criticality in hibernation dynamics" in its current form for publication in Royal Society Open Science.

You can expect to receive a proof of your article in the near future. Please contact the editorial office (openscience@royalsociety.org) and the production office (openscience_proofs@royalsociety.org) to let us know if you are likely to be away from e-mail contact – if you are going to be away, please nominate a co-author (if available) to manage the proofing process, and ensure they are copied into your email to the journal.

on behalf of Dr Cynthia Downs (Associate Editor) and Pete Smith (Subject Editor)
openscience@royalsociety.org

Associate Editor Comments to Author (Dr Cynthia Downs):

Thank you for making the requested changes. In particular, the updated version of Fig 4 helps emphasize the main message of the paper. The changes that were made sufficiently address the reviewers' comments.

Appendix A

Dear Editor,

Thanks for your comments and those from the reviewers. We have addressed all the concerns raised by the reviewers, to whom we have acknowledged his/her help in the new version. We explain below how we have responded to each of the queries on a point-by-point basis (our responses in italics).

All the best,

Daniel Oro

Associate Editor Comments to Author (Dr Cynthia Downs):

Associate Editor: 1

Thank you for submitting to Open Science. Two reviewers and I reviewed this manuscript. This manuscript presents an analysis of the time dynamics of body temperature in two individuals (one male and one female) of Edible dormouse (*Glis glis*). The research shows that the two individuals exhibit a similar pattern of flickering body temperature at the transition between two stable temperature states (hibernation and active). One reviewer provided a favorable review with minimal comments, and one provided more substantial feedback. Reviewer 2 expressed concern that the study used only two dormice and a single year of data. This reviewer suggested that the work could have benefited from a second annual year to better support body temperature patterns entering fall. Although I agree that adding additional individuals would broaden the results' scope, I do not think that this is necessary for publication. Instead, I recommend adding a discussion of the study's limitations that arise from the study design (e.g., sample size and rate of temperature recordings). Additionally, I encourage you to address the comments about the figures and table to help clarify the results and conclusions, particularly figure 4, which is critical to the paper. Specifically, for Fig 4, please add tick marks at the 5's on the y-axis to make it easier to compare body and air temperature.

We have added two sentences at the beginning of the Discussion section highlighting the limitations of our study. We do not understand the issue of the ticks: ticks for the 5's are shown in Y-axis, but not in the right panels on potential, since potential values do not refer to temperatures.

Reviewer comments to Author:

Reviewer: 1

Comments to the Author(s)

This work analyzes the annual cycle of body temperature of the edible dormouse. The study uses sound statistical analysis to show that the transitions to the hibernation period and the converse transition to active live in Spring can be regarded as critical transitions between two alternate resilient (stable) states. In particular, this work shows that such observed transitions have common characteristics (in statistical sense, such as flickering) with the critical transitions close to tipping points between alternate states that have been previously described with the use of mathematical models in other natural systems.

I believe the ms provides a nice example of flickering preceding both forward and backward abrupt transitions between two alternate states. Empirical data also show hysteresis.

The paper uses well-developed statistical tests applied to time series data to detect early warning signals of abrupt transitions. The authors use both metric-based and model-based approaches to detect early warning signals. Model-based methods are not based on a dynamical system where an attempt to include some physiological mechanisms responsible for this alternate behavior could have been explored, but only a statistical time series model.

The work considers outdoor temperature as the control variable, which smoothly changes along the year, driving an abrupt change in the response variable, the body temperature. The main drawback of this work is that there is little discussion about the main physiological mechanisms providing resilience to both alternate states and what causes the erosion of this resilience as the transition is close. This could have been explored with a dynamical system of some sort. The paper uses concepts from dynamical systems, but then only uses a statistical time series model. I believe this exploration was not the goal of the work presented here by these authors. However, some speculative comments in the discussion could have been added.

We have added some speculative comments on these issues in the Discussion as suggested by the Reviewer.

One question to further explore is whether or not there are signs of flickering in air temperature as well. One can imagine that spring and fall are transition seasons (with more temperature fluctuations close to the end of the season) between two alternate states, Summer and Winter.

The Reviewer is right: indeed, we performed such analyses and included them in the original version (see Figure 2). There is no signs of critical transitions or the like in the air temperature time series, so we can assume that the changes in temperature were slow and smooth. To reinforce this idea, we have included a new figure in the Supplementary Material (new Fig. S6) and add either some words or sentences in the Summary, Results and the Discussion to state that air temperature changed smoothly.

Finally, I find Figure 4 central to the paper. I would have added the histogram (as a third column on the left) showing a bimodal distribution of body temperatures around the ones characterizing each of the alternate states.

Since Reviewer#2 also raises some concerns on this figure, I explain here what is new in the revised version: first, I added the histograms where, as suggested by this Reviewer, there is bimodality in the frequency distributions of body temperatures. I have coloured the panels in a coherent manner to facilitate comprehension of the three different figures here. Owing that the lines are drawn not from a dynamical model, which as the Reviewers states is out of the scope of this study, we decided to eliminate the lines and the bifurcation points, which were qualitatively drawn based on the potential analysis values.

Reviewer: 2

Comments to the Author(s)

The authors are providing analyses on the time dynamics of body temperature (T°) in two individuals of the species *Glis glis* in the Mediterranean climate. They use one annual time series of this measure and study critical transitions related to the entrance and exit from hibernation. The authors use several metric-based techniques to identify critical transitions from time series analyses, based on previous literature of Early Warning Signals (EWS, leading indicators). They also use model-based indicators, using e.g., threshold AR(p) models.

In my opinion the authors follow an interesting approach to provide EWS by monitoring an in vivo system. This kind of research is not easy and they did an effort in doing so. They have based on dynamical behaviour with flickering body T° . They have only used two individuals to carry out the analyses, but they show a common pattern with some differences into the entry of the hibernation period. The study would have benefit from using more individuals, not to perform all of the analysis to each of them, but to see whether the signals they find for the body T° remain similar between individuals (or even between ages). The use of more individuals could have been used to extract a common pattern of changes between hibernation and activity, and see the intrinsic variability associated to it under similar patterns of air T° . I pretty much enjoyed the Discussion Section, which is very well written and characterises quite well the topic and work done by the authors. However, a major problem is that the Results Section is very short and poor. The article is well written and novel. However, I have some concerns that should be addressed in a Major revision. Some of these concerns are about the solid evidences they are providing to show that these transitions may occur, especially in the transition towards hibernation.

We believe that the Results section is not so short. It has the same length than the Methods section and it reflect well what is tested in our study. There are several references to the Supplementary Material section, where there are four additional tables and seven additional figures (the previous version had six figures). Furthermore, there is a new sentence related to the analysis of flickering on air temperature.

Major concerns:

The detection of early warning signals (EWS) is easier when the control parameter varies smoothly and the system undergoes an abrupt transition, but the time series are very noisy (air T° is extremely noisy).

[We have broken this first major concern to facilitate our responses] Reviewer# 1 also mentions whether air temperature should show flickering or not. As we noted above, we have added a new sentence in the Results section and a new figure in the Appendix (Fig. S6) about this issue: air temperatures do not show signs of criticality, as it was already shown in the original version (figure 23 –yellow lines).

I have also some questions regarding the experiments and the presentation of some of the results. My main concern is that from the analyses performed (mainly the metric-based measures) I do not see a clear matching between some of the changes in these indicators and the transitions identified in the data (marked with vertical dashed lines in Fig. 1). This is specially important for the entry to hibernation, where the time series starting at time 0 up to the first vertical line is very short.

Please note that even if the period seems short in the context of the whole dataset, it contains more than 1200 data-points for the female and almost 2000 data-points for the male, which implies a very robust sample size to test the different indicators.

I understand that it might have took longer time to do the experiments, but it would have been better to have a two-year time series, as a way to have “temporal” replicates of the transitions for the same individuals. Regarding this concern, for example, the female shows a clear flickering pattern approximately in the middle of the first period (beginning Autumn) towards the first transition, but close to the transition the flickering seems to disappear.

Please note that flickering does occur just prior the transitions even at a simple look in Figure 1 (a). It is true that there is a short period with no flickering, but if you make the exercise to glue around the final part of the figure 1 with the initial part (in a cyclic manner, which is how it really occurs), you may see that flickering occurs not following a neat trend, but it also includes some stochastic behaviour that can be also observed for male data.

For the male this phenomenon is not so evident. In this sense, I am wondering if the title of this article is: “Flickering body temperature anticipates criticality in hibernation dynamics” may be changed. From the words in the title I understand that the main result of the paper is an evident warning signal prior to the entry into hibernation. However, as I previously stated, from the data of the time series shown in Fig. 1, it is not clear to me that the entry into hibernation for the female records a special, and high-frequency, flickering pattern.

Results on EWS indicate so: data on female do show flickering prior to the change of state in the two shifts: to hibernation and to activity. This is why we entitled our work as “hibernation dynamics”, since we refer not only to entering hibernation but also when this ends. We decide not to change the title since Results support the evidence that flickering occurs at the two transitions for the two study individuals.

The quality of the figures is not very good, and, for example, it is difficult to see the data of the air T° in Fig. 1. I would suggest the authors to use high quality figures along the article. I suggest to use another colour (perhaps light grey with some transparency) for the air T° and overlap it to the body T° , and also show the air T° separately. That is, the authors should plot the air T° of the two panels in Fig. 1 alone and separately, perhaps as a Supplementary figure.

We have used the maximum quality available but we can increase dpi definition for final edition if asked. We believe that it is much more useful for the reader to have the two temperatures (body and air) together in the same panel to see how the two vary over time. Colour grey for air temperature does not improve the quality of the figure (we have tried many

other colours but none of them yields a better result). Please note that the number of data points (more than 8000) troubles the identification of single points, but what matters here are the trends over time.

Despite a visual inspection of the body temperature shows the increase in flickering frequency close to the vertical dashed lines of Fig. 1, there exists flickering during all the hibernation period. This means, I guess, that mice have some activity during the hibernation period that becomes more frequent as spring approaches. If this is the case, it should be better explained in the article and in the caption of Fig. 1.

We have added this explanation in the main text (first part of the Results section) and the caption of the figure.

I understand that the body T° was monitored every hour, thus they are not continuous data points. How could have this affected in detecting activity periods shorter than 1 hour? If the time axes in the plots are hours, just write hours.

Body temperature vary with a relatively slow pace. Recording body temperature very hour is the right scale for assessing the temporal changes at the physiological pace of temperature variability. About the x-axis, we have added in the caption of figure 1 that the time refers to hours.

The results Section is extremely short. This section should be worked more (see below).

See our comments earlier and the ones specific to the comments below.

Lines 176-177: the ACF shown in Fig. S1 is large only at lag-1. Makes it sense close to a critical transition? I would expect correlations at several lags.

No, it does not make sense. What is typical in critical transitions where raising memory occurs is an increase in the ACF, which is not our case (as it is explained in the main text). This figure is not particularly interesting, as compared with S4, S5 and S6, which show the behaviour of the autocorrelation function over time and prior to transitions.

Figure 2. Panels (a) and (b) are body temperature? Before hibernation, right? Please put Temperature in the y-axis. Why did you applied a Gaussian filtering to the time series?

Right, we have added this to the upper panels. Panels a and c correspond to the period before hibernation (for the female and the male respectively); panels b and d correspond to the period before activity (for the female and the male respectively). We applied the filter for eliminating the main noise of the time series. We have added an explanation in the Methods section and also in the caption of the figure.

Concerning the period from time = 0 to time = 1800 (approx) - entry into hibernation: The standard deviation (SD) for the female shows a peak at about time = 8000, while the change to the other "hibernation attractor" takes place at about time = 1800. This difference makes me to doubt that this is indicating the presence of a critical transition for this period.

Sorry, we do not get this issue. There is no time 8000, since the time series are split in two periods, prior to hibernation and prior to activity. We do not know either what the Reviewer refers to "hibernation attractor". Yet, we state in the main text that SD does not indicate critical transitions in any case, and none of the other indicators performed better. We have added also more information about this in the Results section.

Also, the SD for the male shows two peaks, thus one could think there are two critical points here (which is not the case, right?). The skewness for the female seems OK, this is increasing. However, the skewness for the mail does not show a clear pattern. You should discuss more this result, and in general, you should discuss more deeply Fig. 2 to convince the reader about your conclusions.

The reviewer is right. We have added a sentence to clarify that there is no a clear pattern in the behaviour of the indicators to warn about critical transitions in body temperatures. We already stated in our Discussion that "Nevertheless, the rest of indicators were less informative and even erratic: for instance, CH did not anticipate bifurcation, while DDJ metrics were very noisy. As Dakos et al. (2012) pointed out when developing the indicators, the performance of any indicator, as well as the interpretations based on them, is likely depending on the features and dynamics of the biological system studied."

Concerning the period from time = 2000 to time = 5800 (approx) - approach to activity and exit from hibernation: The results for the SD are here clearer. However, why is the skewness decreasing? Beyond a more extensive discussion and explanations of the results (including the Suppl. Mat.) I may suggest to include a paragraph at the end of the results Section summarising the EWS indicators confirming the presence of this critical transitions the authors claim to exist. That is, to name the measures allowing to obtain a clear result about the approach to a critical transition.

We have provided in the new version a clearer explanation about the behaviour of the indicators (see our previous response). As stated in previous studies (mainly the one published by Dakos et al. in 2012 about the use of indicators), very little is known about the reliability of

the different indicators, so we provide proof that some of them performed poorly and did not show a pattern in behaviour. We have also added a new sentence in the Summary to highlight this fact.

Figure 3. What does this 31°C threshold for the AR(3) model mean?

It means that this is the threshold the model estimates to separate the two dynamics: the one occurring during the hibernation and the one occurring during activity. We have added a sentence in the Results section for clarification.

Figure 4. Panels (a) and (b) look nice. Here, the two stable states seem more or less clear from the data. I have some questions: have you computed this type of bifurcation diagram with the fitted AR(3) model? If so, you should put this info in the caption of fig. 4. Have you tried to build this diagram using the raw T° data? I would like to see it, perhaps it looks to noisy, but if you plot a hot map with the density of points within given regions you could also observe this “fold”-like behaviour.

As we noted earlier, the bifurcation line was drawn using a qualitative approach from the results obtained in the analysis of the potential (shown also in this figure). To avoid confusion, the new figure does not include this bifurcation diagram, but they add different colours for each basin of attraction and the unstable phase. Analysing the potential for air T° has not sense since we have shown that this variable does not show abrupt changes (figure 2 and S6).

How have the solid lines been placed within this cloud of points? Just visually? Or you did some analyses to make them pass through the more dense cloud of points? I am mentioning it because the lower branches seems ok to me, but the upper ones are missing lots of points at the left part of the plots. The same for the bifurcation points, have been computed or they have drawn visually? How have the potentials of panels (a1) and (b1) been computed?

See our previous response. As we state in our Methods section, we explain how we calculate potentials, and we have added two sentences for clarification.

Figure 5: Why do you talk about hysteresis here? I do not see how you relate hysteresis with the shape of these curves. This plot is very difficult to understand. You are showing two curves (and points), one for hibernation and the other for the euthermic state. But both have the same shape. These results are very difficult to interpret.

Hysteresis occurs when the path from one state to the other is different from the later to the former. Within the framework of resilience, it means that the transition to hibernation does not occur at the air temperature during the transition to activity, but at a lower air temperature. Just googling “hysteresis” the first image you get is:

which is exactly what we show in this figure.

Table S1. Generic EWS: The authors used an overlapping moving window, but, could this affect the autocorrelation measures? They need to justify why not if this is not the case.

This is standard procedure (see Dakos et al. 2012). We use different windows to test the sensitivities of the indicators (see figure S7).

Table S4. The Standard errors are shown by means of the standard deviation, de typical deviation, or the standard error of the mean? These error for parameter ϕ_0 are too large to take conclusions. Where is (A) in the Table?

We have clarified these issues in the caption of the table and in the Table itself. Relative large errors in ϕ do not invalidate our results since they are the intercept parameter in the regression function.

Figure S6. The sensitivities of the generic EWS seem quite heterogeneous and changing to the rolling window metrics. With such a variability I wonder if it is possible to get conclusions from the rolling windows used in the analyses. Am I missing anything?

We note that this figure is now figure S7. As we explain in our Results section, sensitivities of all generic EWS tested in our study were low, i.e. results were robust regardless of different choices on bandwidth and size of the rolling window.

Minor concerns:

The concept of Early Warning Signals (EWS) should be explained in the Introduction.

We have added an explanation in the Introduction. This also highlights the statistical nature of our study, which is noted by the Reviewer #1.

Line 77: critical transitions can also be continuous.

Not in the sense that we state in the sentence.

Line 130: Indicate what does BDS means

Done.

Figure 2. The measures performed in each panel should go in the y-axes. Could you add the symbols of male and female to the panels to distinguish them better? You can do it throughout the paper to ease the interpretation of results for each sex.

We have added the symbols, whereas we keep the y-axis titles within the panels to facilitate their interpretation.

Line 228: Please, write Early Warning Signals (EWS) the first time they appear in the Discussion (or, generically, in a given section).

Done in the new version (in the Introduction, see response above).

Lines 232-233: Please, indicate what CH and DDJ mean.

Even though this is stated in the Methods, we have repeated this in the Discussion as suggested.

Line 255: What does high-dimensional physiological systems mean?

We added an explanation, which also refers to transient dynamics (with a new reference).

Lines 269-271: I think it would be interesting to extend the comments on the consequences of hibernation on population fluctuations and especially, on species extinctions.

We added a further sentence on this topic and two new references. We also added a few word on this issue at the end of the Conclusions section.

Line 306-307: The authors say that flickering occurs when a system rapidly shifts between alternative basins of attraction far from bifurcations. Far or close to bifurcations?

The reviewer is right, we changed this to "close to".